# A comparison of two population-based household surveys in Uganda for assessment of violence against youth

Dustin W. Currie[1,2]*, Rose Apondi[3], Christine A. West[1], Samuel Biraro[4], Lydia N. Wasula[5], Pragna Patel[1], Jennifer Hegle[1], Ashleigh Howard[6,7¤], Regina Benevides de Barros[1], Tonji Durant[1], Laura F. Chiang[6], Andrew C. Voetsch[1], Greta M. Massetti[6]

1 Division of Global HIV & Tuberculosis, Centers for Disease Control and Prevention, Atlanta, Georgia, United States of America, 2 Epidemic Intelligence Service, Centers for Disease Control and Prevention, Atlanta, Georgia, United States of America, 3 Division of Global HIV & Tuberculosis, Centers for Disease Control and Prevention, Kampala, Uganda, 4 ICAP at Columbia University, Kampala, Uganda, 5 Uganda Ministry of Gender, Labour, and Social Development, Kampala, Uganda, 6 Division of Violence Prevention, Centers for Disease Control and Prevention, Atlanta, Georgia, United States of America, 7 Strategic Innovative Solutions, LLC, Clearwater, Florida, United States of America

¤ Current address: Division of Global Health Protection, Centers for Disease Control and Prevention, Kinshasa, Democratic Republic of the Congo
* pif7@cdc.gov

**Data Availability Statement:** UPHIA data are publicly available at the following URL: https://phia-data.icap.columbia.edu/datasets?country_id=7. Uganda VACS data are available through the

## Abstract

Violence is associated with health-risk behaviors, potentially contributing to gender-related HIV incidence disparities in sub-Saharan Africa. Previous research has demonstrated that violence, gender, and HIV are linked via complex mechanisms that may be direct, such as through forced sex, or indirect, such as an inability to negotiate safe sex. Accurately estimating violence prevalence and its association with HIV is critical in monitoring programmatic efforts to reduce both violence and HIV. We compared prevalence estimates of violence in youth aged 15–24 years from two Ugandan population-based cross-sectional household surveys (Uganda Violence Against Children Survey 2015 [VACS] and Uganda Population-based HIV Impact Assessment 2016–2017 [UPHIA]), stratified by gender. UPHIA violence estimates were consistently lower than VACS estimates, including lifetime physical violence, recent intimate partner physical violence, and lifetime sexual violence, likely reflecting underestimation of violence in UPHIA. Multiple factors likely contributed to these differences, including the survey objectives, interviewer training, and questionnaire structure. VACS may be better suited to estimate distal determinants of HIV acquisition for youth (including experience of violence) than UPHIA, which is crucial for monitoring progress toward HIV epidemic control.

## Introduction

Violence against children (ages 14 years and under) and youth (ages 15–24 years) [1], defined by the World Health Organization as "the intentional use of physical force or power,

following URL: https://www.togetherforgirls.org/request-access-vacs/.

**Funding:** This project has been supported by the President's Emergency Plan for AIDS Relief (PEPFAR) through the Centers for Disease Control and Prevention and U.S. Agency for International Development under the terms of cooperative agreements #U2GGH001226, GH001619, and GGH000466; UNICEF; and Wellspring Foundation. The findings and conclusions in this report are those of the authors and do not necessarily represent the official position of the funding agencies.

**Competing interests:** The authors have declared that no competing interests exist.

threatened or actual, against oneself, another person, or a group or community, that either results in or has a high likelihood of resulting in injury, death, psychological harm, maldevelopment, or deprivation" [2], is a significant public health issue: over one billion children are affected each year globally [3]. Individuals who experience violence face wide-ranging consequences, including poor physical and mental health, decreased education and employment opportunities, higher likelihood of drinking alcohol or using drugs in general and before sex, initiating intercourse at an earlier age, having intercourse with multiple partners or strangers, having a sexually transmitted infection, and experiencing suicidal ideation [4–6].

Sub-Saharan Africa has among the highest global prevalence rates of violence against children, youth and women [3,7]. The Sub-Saharan Africa region also has the highest global prevalence and incidence of HIV; 64% of all people living with HIV live in sub-Saharan Africa [8]. Women and girls account for 63% of all new HIV infections in the region, which rises to 86% in the 15–19 year old age group [9]. In Uganda, an estimated 22% of women aged 15 to 49 had experienced some form of sexual violence in their lifetime [10], and an estimated 7.6% of females aged 15 to 64 were living with HIV [11]. Previous studies have suggested an association between violence and HIV risk, through direct mechanisms such as forced sex or indirect mechanisms such as an inability to negotiate safe sex [12–14]. Gender and violence are often intertwined, and the higher incidence of HIV among 15–24 year old females compared to their male peers in some countries likely reflect gender-related drivers of HIV infection, including violence [12,14–16].

National household surveys funded by the US President's Emergency Plan for AIDS Relief in sub-Saharan Africa have been designed to estimate the prevalence of violence in relation to sexual risk behaviors and HIV status. Violence Against Children and Youth Surveys (VACS; https://www.cdc.gov/violenceprevention/childabuseandneglect/vacs/) are designed to estimate childhood, lifetime, and past-12-month violence prevalence in children and youth aged 13–24 years and include modules assessing sexual behaviors and HIV risk [17]. Population-based HIV Impact Assessments (PHIAs; https://phia.icap.columbia.edu/) are designed to estimate HIV-related outcomes and in most countries included a violence module during the first round of surveys; most countries use an abbreviated version to allow for inclusion of other priority issues.

The Uganda PHIA 2016–2017 (UPHIA) used the full violence module and thus is aligned more closely with the Uganda VACS (2015). UPHIA eligibility criteria were adapted to be comparable with VACS criteria, by restricting violence module eligibility to youth and by modeling the sampling approach after VACS [11]. Therefore, UPHIA served as a "proof of concept", allowing determination of whether an HIV-focused survey with biomarker data collection resulted in comparable violence prevalence data to VACS. The purpose of this report is to compare violence prevalence estimates generated by VACS and UPHIA. Understanding comparability of violence prevalence estimates in an HIV-focused survey and a violence-focused survey has implications for ongoing surveillance of violence against youth and HIV.

## Methods

Detailed descriptions of the VACS [17,18] and UPHIA [11] survey designs have been published previously. Briefly, both were population-based household surveys using a three-stage sampling approach, in which enumeration areas (EAs), limited geographic areas designated by national statistical authorities, were selected within regional strata in Uganda, and households were randomly selected within EAs. A single eligible respondent per household was selected randomly to participate in the survey (for VACS) or the violence module (for UPHIA). For both surveys, a referral mechanism was in place to put respondents in contact with a social

**Table 1. Methods and sampling structure of Uganda Population-based HIV Impact Assessment (UPHIA 2016–2017) and Uganda Violence Against Children Survey (VACS 2015).**

| Survey Design Element | VACS | UPHIA |
|---|---|---|
| Data Collection Timeframe | September–November 2015 | August 2016–March 2017 |
| Primary Objective | Estimate national prevalence of violence among children and youth | Estimating national-level annual HIV incidence among adults and national and subnational prevalence of HIV and HIV viral load suppression among HIV-positive adults |
| Violence Questionnaire Eligibility | Males and females aged 13–24 years | Males and females aged 15–24 years[†] |
| Enumeration Area (EA) Sampling | Splits EAs into female or male (only one sex is sampled within each EA) | Splits EAs into female or male (only one sex is sampled within each EA[†]) |
| Eligibility within Households | Interviewed only one participant per household | Only one violence module completed per household; the rest of the interview completed with all eligible household members |
| Survey Administration Method | In-person, face-to-face | In-person, face-to-face |
| Consent Process | Multi-tiered consent process | Multi-tiered consent process; no additional consent for violence module |
| Interviewer Training | Interviewers trained in building rapport with adolescents and young adults and violence-specific data collection | Interviewers trained in building rapport among participants and service referral when necessary, less emphasis on violence data collection specifically or interviewing adolescents and young adults |
| Interviewer Sex | Interviewers of participant's same sex | Interviewers of participant's same sex, when possible |
| Response Plan | Detailed response plan for participants who needed and wanted help. When possible, on-call social workers were contacted by the interviewer while still in the home for immediate counseling and coordination. Otherwise the social worker would make contact with the participant within 72 hours for counseling services and additional referrals. | Response plan outlined in an SOP; interviewers were instructed to provide referrals with follow-up for any participant who met criteria and consent to referral or who requested services |
| Sample Size[†] | 5,804 (males, 2,645; females, 3,159) | 4,069 (males, 1,762; females, 2,307) |

Abbreviations: SOP, standard operating procedure.

[†]For UPHIA 2016–2017, refers specifically to the violence module and not to the larger PHIA survey.

welfare officer if further assistance was needed for a violence-related issue. National weights in both surveys were used to generate estimates that accounted for sample selection probabilities and were adjusted for nonresponse and noncoverage. Both surveys were approved by the CDC Institutional Review Board (Protocols #6538 [Uganda VACS] and #6830 [UPHIA]), and the Uganda National Council for Science and Technology.

Tables 1 and 2 summarize key differences between the two surveys. While the overarching strategy was similar, there were differences in primary objectives, survey methods and sampling (Table 1), and in questionnaire structure and content (Table 2). Both surveys were designed to generate nationally representative estimates of primary outcomes, and separated EAs such that only males or only females were eligible for either the violence module (UPHIA) or the whole questionnaire (Uganda VACS) within an EA (Table 1). Both surveys were completed via face-to-face interviews with trained interviewers but given the difference in the objectives of the two surveys, there were some differences in interviewer training. There were differences in how experience of violence questions were asked, with UPHIA generally using more parent questions and skip patterns, and Uganda VACS asking about experiences of different types of violence via different perpetrators separately (Table 2).

**Table 2. Wording and response options of comparable violence questionnaire items between Uganda Violence Against Children Survey (VACS 2015) and Uganda Population-based HIV Impact Assessment (UPHIA 2016–2017)** [*].

| Indicator | VACS 2015 | VACS Responses | UPHIA 2016–2017 | PHIA Responses |
|---|---|---|---|---|
| **Physical Violence (Lifetime)** | Has (a romantic partner) ever punched, kicked, whipped, or beat you with an object? Has (a person your own age) ever punched, kicked, whipped, or beat you with an object? Has (a parent, adult caregiver, or other adult relative) ever punched, kicked, whipped, or beat you with an object? Has (an adult in the community, such as teachers, police, employers, religious leader, etc.) ever punched, kicked, whipped, or beat you with an object? | Individual item responses: 1 –Yes 2 –No 99 –Don't Know/ Declined Collapsed into single item: 1 –Yes to any physical violence by any perpetrator 2 –No physical violence reported | Has anyone ever done any of these things to you: • Punched, kicked, whipped, or beat you with an object, • Slapped you, threw something at you that could hurt you, pushed you or shoved you • Choked, smothered, tried to drown you, or burned you intentionally • Used or threatened you with a knife, gun, or other weapon? (asked in single question) | 1 –Yes 2 –No • 8 –Don't Know • 9 –Refused |
| | Has (a romantic partner) ever strangled, suffocated, tried to drown you, or burned you intentionally? Has (a person your own age) ever strangled, suffocated, tried to drown you, or burned you intentionally? Has (a parent, adult caregiver, or other adult relative) ever strangled, suffocated, tried to drown you, or burned you intentionally? Has (an adult in the community, such as teachers, police, employers, religious leader, etc.) ever strangled, suffocated, tried to drown you, or burned you intentionally? | | | |
| | Has (a romantic partner) ever used or threatened you with a knife, gun, or other weapon? Has (a person your own age) ever used or threatened you with a knife, gun, or other weapon? Has (a parent, adult caregiver, or other adult relative) ever used or threatened you with a knife, gun, or other weapon? Has (an adult in the community, such as teachers, police, employers, religious leader, etc.) ever used or threatened you with a knife, gun, or other weapon? | | | |
| **Physical Intimate Partner Violence (Past Year)** [†] | In the last 12 months, has a romantic partner punched, kicked, whipped, or beat you with an object? | 1 –Yes 2 –No 99 –Don't Know/ Declined | If yes to physical violence (lifetime) question above AND reported somebody doing this to them in the past 12 months: In the last 12 months, did a partner do any of these things to you? | 1 –One or more times in past 12 months 2 –No physical violence ever OR No physical intimate partner violence ever OR zero times in past 12 months |
| | In the last 12 months, has a romantic partner strangled, suffocated, tried to drown you, or burned you intentionally? | 1 –Yes 2 –No 99 –Don't Know/ Declined | | |
| | In the last 12 months, has a romantic partner used or threatened you with a knife, gun, or other weapon? | 1 –Yes 2 –No 99 –Don't Know/ Declined | | |
| **Any Sexual Abuse (Lifetime)** | Indicator of whether any of the following four survey items were reported (unwanted sexual touching, attempted forced sex, physically forced sex, and pressured into sex) | 1 –Yes 2 –No | Indicator of whether any of the following four survey items were reported (unwanted sexual touching, attempted forced sex, physically forced sex, and pressured into sex) | 1 –Yes 2 –No |

(*Continued*)

**Table 2.** (Continued)

| Indicator | VACS 2015 | VACS Responses | UPHIA 2016–2017 | PHIA Responses |
|---|---|---|---|---|
| **Unwanted Sexual Touching (Lifetime)** [*†] | Has anyone ever touched you in a sexual way without you wanting to but did not try and force you to have sex? Touching in a sexual way without permission includes fondling, pinching, grabbing, or touching you on or around your sexual body parts. | 1 –Yes<br>2 –No<br>99 –Don't Know/ Declined | How many times has anyone ever touched you in a sexual way without your permission but did not try and force you to have sex? Touching in a sexual way without permission includes fondling, pinching, grabbing, or touching you on or around your sexual body parts. | 1–1 or more times2 –Zero times<br>• 8 –Don't Know<br>• 9 –Refused<br>(Recoded) |
| **Attempted Forced Sex (Lifetime)** [*†] | Has anyone ever tried to make you have sex against your will but did not succeed? | 1 –Yes<br>2 –No<br>99 –Don't Know/ Declined | How many times in your life has anyone tried to make you have sex against your will but did not succeed? | 1–1 or more times2 –Zero times<br>• 8 –Don't Know<br>• 9 –Refused |
| **Physically Forced Sex (Lifetime)** [*†] | Has anyone ever physically forced you to have sex and did succeed? | 1 –Yes<br>2 –No<br>99 –Don't Know/ Declined | How many times in your life have you been physically forced to have sex? | 1–1 or more times2 –Zero times<br>• 8 –Don't Know<br>• 9 –Refused |
| **Pressured into Sex** [*†] | Has anyone ever pressured you to have sex, through harassment, threats, or tricks and did succeed? | 1 –Yes<br>2 –No<br>99 –Don't Know/ Declined | How many times in your life has someone pressured you to have sex through harassment, threats, and tricks but without force and did succeed?<br>Being pressured can include being worn down by someone who repeatedly asks for sex, feeling pressured by being lied to, being told promises that were untrue, having someone threaten to end a relationship or spread rumors or sexual pressure due to someone using their influence or authority. | 1–1 or more times2 –Zero times<br>• 8 –Don't Know<br>• 9 –Refused |

[*]Emotional violence comparisons are not made in this report because VACS asked only about emotional violence by a parent or caregiver, whereas UPHIA asked about emotional violence by any person.

[†]Limited to those who have ever been married or partnered in both surveys.

[*†]The original sexual violence items in UPHIA had multiple response options with 0 = zero times, 1 = 1 to 5 times, and 2 = 5 or more times. These variables were recoded by collapsing responses 1 and 2 into a single value to create an indicator variable with 1 = at least 1 time and 0 = zero times.

Weighted prevalence estimates and 95% confidence intervals (CI) were calculated stratified by gender, accounting for the complex sampling design of each survey. Comparisons between UPHIA and VACS were restricted to individuals aged 15–24 years so that comparisons included identical age groups. Two proportion z-tests were used to calculate p-values comparing prevalence estimates between VACS and UPHIA, with statistical significance determined at $p < 0.05$. Analyses were completed using SAS 9.4.

## Results

Demographic characteristics of the two groups were comparable after weighting (Table 3). Slightly over half of participants were female (UPHIA, 51.3%; VACS, 52.6%), with similar mean age distributions (UPHIA males, 19.1 years; VACS males, 18.9 years; UPHIA females, 19.2 years; VACS females, 19.4 years). Fewer male than female participants had been married or lived with someone like they were married (UPHIA males, 19.3%; VACS males, 19.5%; UPHIA females, 45.2%; VACS females, 46.9%) and had been previously told they were HIV-positive (UPHIA males, 0.1%; VACS males, 0.3%; UPHIA females, 1.6%; VACS females, 1.7%). None of the sex-specific differences between UPHIA and VACS presented in Table 3 were statistically significant ($p > 0.05$).

Lifetime prevalence of physical violence was significantly higher in VACS than UPHIA for both males (VACS, 77.1%; UPHIA, 32.0%; $p < 0.0001$; Fig 1) and females (VACS, 67.0%;

**Table 3. Weighted demographic characteristics and self-reported HIV status of Uganda VACS 2015 and UPHIA 2016–17 participants, by gender.**

| | Uganda VACS 2015 | | UPHIA 2016–17 | |
|---|---|---|---|---|
| | **Females** | **Males** | **Females** | **Males** |
| | **N = 3,159 (52.6%)\*** | **N = 2,645 (47.4%)\*** | **N = 2,307 (51.3%)** | **N = 1,762 (48.7%)** |
| Age in years [Mean (95% Confidence interval)] | 19.4 years (19.2–19.6 years) | 18.9 years (18.7–19.1) | 19.2 years (19.1–19.3) | 19.1 years (19.0–19.2) |
| Marital Status\* | | | | |
| Ever Married/like married | 1259 (46.9%) | 411 (19.5%) | 1190 (45.2%) | 397 (19.3%) |
| Never Married | 1276 (53.1%) | 1653 (80.5%) | 1116 (54.8%) | 1365 (80.7%) |
| Self-report HIV Status\* | | | | |
| 'HIV-positive | 39 (1.7%) | 9 (0.3%) | 32 (1.6%) | 2 (0.1%) |
| Not self-report HIV-positive | 2438 (98.3%) | 1978 (99.7%) | 2242 (98.4%) | 1739 (99.9%) |

\*Presented as number of participants (weighted %). Marital status missing for 3 females in Uganda VACS and 1 female for UPHIA. Self-report HIV status missing for 60 females and 77 males in Uganda VACS, and 32 females and 21 males in UPHIA.

UPHIA, 26.0% p<0.0001; Fig 2). Similarly, past-year prevalence of physical intimate partner violence was significantly higher in VACS for females (VACS, 14.7%; UPHIA, 7.8%; p<0.0001) and males (VACS, 4.4%; UPHIA, 1.6%; p = 0.003).

Prevalence estimates of sexual violence between UPHIA and VACS were more comparable, but some differences in estimates emerged. For females, lifetime sexual violence was significantly higher in VACS (48.6%) than UPHIA (36.2%; p<0.001), as was unwanted sexual touching (VACS: 35.5%; UPHIA: 26.0%; p<0.0001) and attempted forced sex (VACS: 25.0%; UPHIA: 19.5%; p = 0.01). Meanwhile, being pressured to have sex was significantly higher in UPHIA (10.0%) than in VACS (6.6%; p = 0.006). For males, lifetime sexual violence was significantly higher in VACS (26.3%) than UPHIA (22.6%; p = 0.03), while forced sex was significantly higher in UPHIA (7.1%) than in VACS (3.1%; p<0.0001).

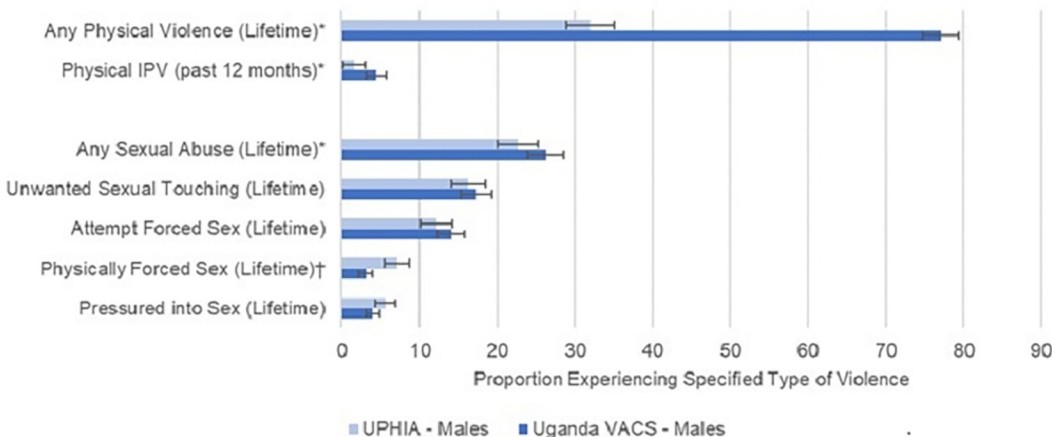

**Fig 1. Lifetime and past-12-month prevalence of sexual and physical violence in males aged 15–24 years in Uganda Population-based HIV Impact Assessment (UPHIA 2016–2017) and Uganda Violence Against Children Survey (VACS 2015).** Violence domains include lifetime physical violence, physical intimate partner violence (IPV) in the past 12 months, and lifetime sexual violence. Asterisks (\*) indicate significantly higher prevalence in VACS than UPHIA (p<0.05). Daggers (†) indicate significantly higher prevalence in UHPIA than VACS (p<0.05).

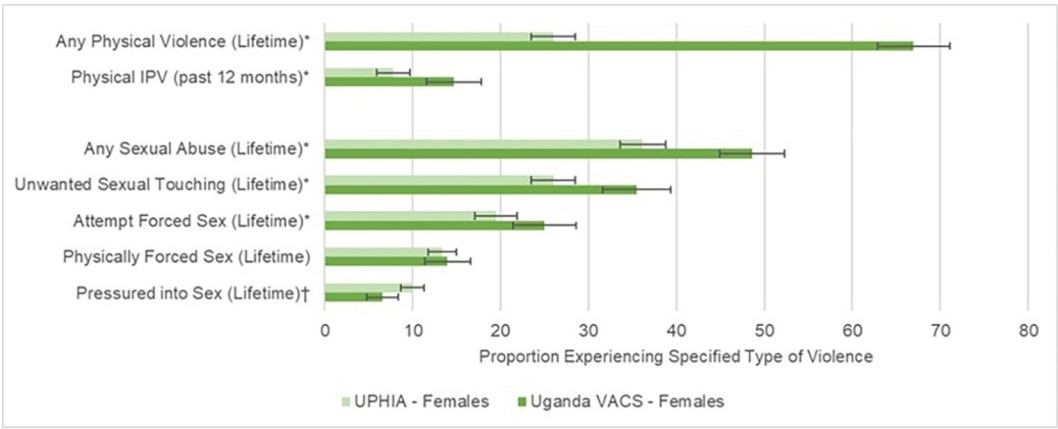

**Fig 2. Lifetime and past-12-month prevalence of sexual and physical violence in females aged 15–24 years in Uganda Population-based HIV Impact Assessment (UPHIA 2016–2017) and Uganda Violence Against Children Survey (VACS 2015).** Violence domains include lifetime physical violence, physical intimate partner violence (IPV) in the past 12 months, and lifetime sexual violence. Asterisks (*) indicate significantly higher prevalence in VACS than UPHIA (p<0.05). Daggers (†) indicate significantly higher prevalence in UHPIA than VACS (p<0.05).

## Discussion

Compared to VACS, UPHIA generated significantly lower prevalence estimates of lifetime physical violence, any lifetime sexual violence, and past-12-month intimate partner violence among both males and females. The differences between survey findings were more consistent for physical violence. Findings were mixed in domain-specific sexual abuse analyses; among males, lifetime reports of having been physically forced to have sex were higher in UPHIA, and among females, being pressured into having sex was higher in UPHIA while unwanted sexual touching and attempted forced sex were higher in VACS. Since underreporting is much more likely in violence data collection than overreporting [19], the overall findings suggest that UPHIA likely underestimated the burden of violence against youth, particularly physical violence, in Uganda [19,20].

Methodological differences between the two surveys likely contributed to these differences. VACS was designed specifically to estimate violence prevalence, while UPHIA was designed primarily to estimate HIV-related outcomes with an additional module to measure violence as a secondary aim. Adding a violence module to a larger survey focused on other health issues may result in underestimation of violence [21]. This can be attributed to confidentiality concerns, survey fatigue, and discomfort with the nature of questions related to violence when compared to other survey items [19]. Additionally, interviewer selection and training related to violence data collection are crucial; respondent willingness to disclose violence may be affected by the rapport established between the interviewer and respondent [21–23]. Interviewers may exhibit implicit biases, either perceived or real, that could result in underreporting. Although substantial interviewer training was conducted for data collectors in both UPHIA and VACS, the VACS interviewer selection and training focused more specifically on strategies to facilitate disclosure and assess violence in the targeted age group [24].

The content and structure of the questionnaire also likely contributed to differences in prevalence across the surveys. For physical violence, the VACS questionnaire repeats stem questions for each of the four categories of perpetrators (intimate partners, peers, parents/caregivers/other adult relatives, and adults in the community) to anchor the responses to different types of relationships and facilitate recall. UPHIA asked one question about physical violence from any perpetrator and followed affirmative answers with a question about the perpetrator.

Given the normalization of many forms of violence, without orienting respondents to the possible perpetrator, they may not consider certain acts to be violent (e.g., violence perpetrated by a peer or a caregiver) [25]. The respondent may think of only the most salient relationships in their life rather than thinking about all of the types of relationships addressed in the VACS questionnaire. Conversely, UPHIA provided additional clarification to the respondent regarding examples of being pressured into sex, which was the only violence domain more frequently reported in UPHIA than VACS among females. Previous research has demonstrated that using multiple behaviorally specific questions about violence generates higher prevalence estimates than broader, aggregate questions [25]. Our findings suggest that the structure of violence questionnaires, specificity of questions, and design of parent questions and skip patterns may be critical in avoiding under-reporting of violence. Table 2 demonstrates that the structure of the UPHIA questionnaire for physical violence was much different than the VACS, while they were more similar for sexual violence; underestimates for physical violence in UPHIA were also more pronounced than for sexual violence.

PHIAs provide valid and reliable information on important indicators of HIV epidemic control, including incidence, prevalence, and progress toward Joint United Nations Programme on HIV/AIDS 95-95-95 targets (95% of HIV-positive individuals are aware of their HIV status; of these, 95% are receiving antiretroviral therapy; and of these, 95% have achieved viral load suppression) in countries with generalized HIV epidemics. However, our findings suggest that enhancing the VACS to include key laboratory outcomes relating to HIV (e.g. HIV testing, incidence testing, and viral load testing) rather than incorporating additional violence data collection into HIV-focused PHIAs, may be a better approach to understand the prevalence of violence in youth and the relationship between violence and HIV. This approach has been implemented to varying degrees in several recent VACS, which also included rapid HIV testing (in Botswana, Côte d'Ivoire, Lesotho, Kenya, Namibia, Zimbabwe, and Mozambique); the violence module of PHIA has been removed for all countries as of 2019. UPHIA's violence module represented an attempt to incorporate lessons learned from VACS surveys to estimate violence prevalence in an HIV-focused survey; however, many violence-related outcomes were underestimated in UPHIA. Generating reliable violence estimates in an HIV-focused survey may be possible, given the similar findings in some of the similarly worded sexual violence questions between VACS and UPHIA. However, this would likely require further changes to questionnaire structure (further increasing interview length), as well as additional specialized training for data collectors in facilitating violence disclosure. VACS and PHIA have different primary objectives, and VACS may be better suited to estimate distal determinants of HIV acquisition for youth (including experience of violence), since they can focus more of the interview time as well as the recruiting and training of the interviewers on these topics.

## Acknowledgments

We thank the Uganda Ministry of Health and Ministry of Gender, Labour, and Social Development for their leadership in carrying out each of the surveys, the many staff who collected data for both surveys, and the participants of both surveys for sharing personal details of their lives with survey staff, and without whom none of this work would be possible.

## Author Contributions

**Conceptualization:** Dustin W. Currie, Rose Apondi, Christine A. West, Lydia N. Wasula, Pragna Patel, Jennifer Hegle, Ashleigh Howard, Regina Benevides de Barros, Tonji Durant, Andrew C. Voetsch, Greta M. Massetti.

**Data curation:** Rose Apondi, Samuel Biraro, Lydia N. Wasula, Ashleigh Howard, Laura F. Chiang.

**Formal analysis:** Dustin W. Currie, Ashleigh Howard.

**Funding acquisition:** Christine A. West, Andrew C. Voetsch.

**Methodology:** Dustin W. Currie, Rose Apondi, Pragna Patel, Jennifer Hegle, Ashleigh Howard, Regina Benevides de Barros, Laura F. Chiang, Andrew C. Voetsch, Greta M. Massetti.

**Project administration:** Christine A. West, Samuel Biraro, Lydia N. Wasula, Ashleigh Howard, Andrew C. Voetsch.

**Resources:** Pragna Patel, Jennifer Hegle, Ashleigh Howard, Regina Benevides de Barros, Tonji Durant.

**Supervision:** Christine A. West, Andrew C. Voetsch, Greta M. Massetti.

**Writing – original draft:** Dustin W. Currie.

**Writing – review & editing:** Rose Apondi, Christine A. West, Samuel Biraro, Lydia N. Wasula, Pragna Patel, Jennifer Hegle, Ashleigh Howard, Regina Benevides de Barros, Tonji Durant, Laura F. Chiang, Andrew C. Voetsch, Greta M. Massetti.

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
