## [Decision Letter · Decision Letter 0]

12 Oct 2021

PONE-D-21-25230A comparison of two population-based household surveys in Uganda for assessment of violence against young adultsPLOS ONE

Dear Dr. Currie,

Thank you for submitting your manuscript to PLOS ONE. After careful consideration, we feel that it has merit but does not fully meet PLOS ONE’s publication criteria as it currently stands. Therefore, we invite you to submit a revised version of the manuscript that addresses the points raised during the review process.

We look forward to receiving your revised manuscript.

Kind regards,

Lindsay Stark

Academic Editor

PLOS ONE

Journal Requirements:

Reviewers' comments:

Reviewer's Responses to Questions

**Comments to the Author**

1. Is the manuscript technically sound, and do the data support the conclusions?

Reviewer #1: Yes

Reviewer #2: Yes

2. Has the statistical analysis been performed appropriately and rigorously? 

Reviewer #1: Yes

Reviewer #2: Yes

3. Have the authors made all data underlying the findings in their manuscript fully available?

Reviewer #1: Yes

Reviewer #2: Yes

4. Is the manuscript presented in an intelligible fashion and written in standard English?

Reviewer #1: Yes

Reviewer #2: Yes

5. Review Comments to the Author

Reviewer #1: Dear Editor,

Thank you for the opportunity to review the manuscript titled “A comparison of two population-based household surveys in Uganda for assessment of violence against young adults”. This paper does an important need – advancing our understanding of the comparability of household datasets that serve as the primary source of data that is the basis of many programmatic and policy decisions, particularly on issues such as violence against children that are vastly under researched. The manuscript is well written; the research objectives are clearly articulated; and the utilizes sound methodology.

I have one comment about the manuscript. The authors’ attempt to provide guidance on how best to address the discrepancies identified is laudable but I am confused by the suggestion that “our findings suggest that enhancing the HIV module of 178 VACS with biomarker data (e.g., incorporating HIV testing), rather than incorporating additional 179 violence data collection into HIV-focused PHIAs, may be a better approach to understand the 180 prevalence of violence in youth and the relationship between violence and HIV.” How would including biomarker data on HIV testing advance this goal? This would be understandable if the authors were suggesting biomarkers of chronic stress? Also, I understand that household surveys have several constraints – the budget, survey lengths, country priorities etc., should we not consider perhaps adapting strategies that have been effective for the VACS? Given that VACS does not include HIV related data, I would think that strengthening this component within a HIV focused survey would most likely advance our goals of understanding the consequences of violence in this population.

Reviewer #2: The is manuscript covers an important topic and the findings can provide a useful contribution to the measurement of violence against children and young adults. Between two studies, VACS and UPHIA, the authors find significant differences in violence prevalence for both females and males aged 15-24 in Uganda. The discussion provides practical implications for measuring violence against children and young adults within HIV studies.

Abstract

- The authors mention that the association of violence to health-risk behaviors may contribute to gender-related HIV disparities. While this is an important point, it is also important to consider that sexual violence may directly contribute to HIV transmission. The abstract would be more compelling if it started by acknowledging the complex ways in which violence, gender, and HIV are linked.

Introduction

- Lines 40-42: The source of the definition for violence against children and youth is unclear. The citation provided is relevant for the annual estimation that over one billion children are affected by violence globally; however, the reference does not provide the quoted definition.

- Lines 44-46: It would strengthen the authors' introduction to amend this sentence to describe the wide-ranging consequences faced by individuals who experience violence during childhood per the Berenson, Wiemann, and McCombs article and other VAC-specific articles.

Lines 46-47: The sentence that begins with "In addition to having the highest HIV prevalence..." starts quite abruptly. The authors may want to consider including a separate paragraph to describe the violence and HIV risks in sub-Saharan Africa or Uganda specifically.

- It is not always clear what differences exist when the authors mention children, youth, young adults, or adolescents. It would strengthen the readability if the authors used consistent terminology and provided corresponding definitions when utilizing various age-specific terminology.

Methodology

- The authors provide a clear description of the similarities between the VACS and UPHIA implementation and the authors guide readers to Tables 1 and 2 to better understand differences. It is important that the authors add a paragraph describing key differences as well.

- When discussing the similarities between the VACS and UPHIA sampling strategies, it would be helpful to mention the split-stage design utilized by both.

- Table 2: There seem to be a few formatting inconsistencies or errors that the authors should review. For example, the narrative regarding repeated questioning per perpetrator is described differently in the first cell than in the second and third cell. As another example, the first example of "Physical Intimate Partner Violence (Past Year)" is an example of lifetime experience: "Has a romantic partner ever punched, kicked, whipped, or beat you with an object."

Results

- Lines 106-112: The content in the first paragraph would benefit from a complementary Table.

- Lines 106-113: It is unclear where there are statistically significant differences in findings provided in this paragraph, which is important since the authors describe differences and similarities between the two studies.

- Lines 108-109: It would help if the authors provided CIs rather than the IQRs

Discussion

- Lines 141-143: The authors describe "significantly lower prevalence estimates of both lifetime physical and sexual violence for both males and females"; however, the findings related to sexual violence are mixed. For some questions, the UPHIA resulted in higher prevalence.

- Lines 142-143: The authors describe that past 12-month IPV was lower in the UPHIA than the VACS for females. It is unclear why the authors did not mention that past 12-month physical IPV was lower in the UPHIA than the VACS for males, as described in lines 116-117.

- Lines 143: Suggest that the authors use the term "more consistent" rather than "largest"

- Lines 153-158: Sexual violence is often more sensitive to disclose; yet, there was not an overall lower reporting in the UPHIA than the VACS. The authors should clarify how interviewer selection and training may have influenced reporting for physical violence but not sexual violence.

- Lines 159-172: The discussion would benefit from the authors further describing why they think the VACS consistently measured higher prevalence of physical violence but not sexual violence. Lines 168-172 indicate that the specificity of questions, including forms of violence and specific perpetrators, may be critical to increased reporting.

- When similar questions were asked in the VACS and UPHIA, such as sexual violence questions, results were more consistent between the two studies. Thus, the authors opinion that the PHIAs should not incorporate additional violence questions seems inconsistent with the evidence. However, the suggestion for the VACS to include biomarker data is compelling.

- Lines 184-185: The evidence provided in this manuscript suggests that the inclusion of perpetrators within the physical violence questions of the VACS enables it to better measure violence as a distal determinant of HIV acquisition, but the authors should clarify if/what other factors contribute to this conclusion.

6. PLOS authors have the option to publish the peer review history of their article (what does this mean?). If published, this will include your full peer review and any attached files.

Reviewer #1: No

Reviewer #2: No

---

## [Author Response · Author response to Decision Letter 0]

18 Nov 2021

Reviewer #1: Dear Editor,

Thank you for the opportunity to review the manuscript titled “A comparison of two population-based household surveys in Uganda for assessment of violence against young adults”. This paper does an important need – advancing our understanding of the comparability of household datasets that serve as the primary source of data that is the basis of many programmatic and policy decisions, particularly on issues such as violence against children that are vastly under researched. The manuscript is well written; the research objectives are clearly articulated; and the utilizes sound methodology.

We would like to thank the reviewer for their thoughtful review of our manuscript.

I have one comment about the manuscript. The authors’ attempt to provide guidance on how best to address the discrepancies identified is laudable but I am confused by the suggestion that “our findings suggest that enhancing the HIV module of 178 VACS with biomarker data (e.g., incorporating HIV testing), rather than incorporating additional 179 violence data collection into HIV-focused PHIAs, may be a better approach to understand the 180 prevalence of violence in youth and the relationship between violence and HIV.” How would including biomarker data on HIV testing advance this goal? This would be understandable if the authors were suggesting biomarkers of chronic stress? 

We understand the confusion with the biomarker language – this is terminology that we use broadly in PHIAs as an umbrella term for laboratory related HIV outcomes (e.g., antibody testing, incidence testing, viral load, ARV metabolite detection, etc.). We have revised the language to more clearly reflect this in the manuscript.

Also, I understand that household surveys have several constraints – the budget, survey lengths, country priorities etc., should we not consider perhaps adapting strategies that have been effective for the VACS? Given that VACS does not include HIV related data, I would think that strengthening this component within a HIV focused survey would most likely advance our goals of understanding the consequences of violence in this population.

Thank you for raising this important point. We agree with the reviewer that household surveys, particularly those focused on HIV, have the constraints acknowledged (budget, survey length and corresponding participant response rates and willingness to participate, country priorities, etc.). VACS is also a household survey, but has the benefit of focusing specifically on violence, which appears to elicit more reliable violence prevalence data. UPHIA was an attempt to do just what the reviewer is suggesting, strengthening an HIV focused survey to collect violence data using lessons learned from VACS. Unfortunately, UPHIA resulted in underestimates, as described in the study. Therefore, we think the approach of including HIV related laboratory data in VACS (which is already underway in some surveys) may be more beneficial. We’ve attempted to make this point more explicitly in lines 218-227 of the revised manuscript. We’ve also acknowledged in the revised manuscript that it may yet be possible to collect reliable violence estimates in an HIV-focused household survey, but that further changes to the approach would be required. These further changes may jeopardize the ability of PHIAs to achieve their primary objectives, and so the violence module was removed in PHIAs in favor of continuing to complete VACS surveys for the purposes of violence estimation (in adolescent girls and boys and young women and men).

Reviewer #2: The is manuscript covers an important topic and the findings can provide a useful contribution to the measurement of violence against children and young adults. Between two studies, VACS and UPHIA, the authors find significant differences in violence prevalence for both females and males aged 15-24 in Uganda. The discussion provides practical implications for measuring violence against children and young adults within HIV studies.

Thank you very much for your thoughtful review.

Abstract

- The authors mention that the association of violence to health-risk behaviors may contribute to gender-related HIV disparities. While this is an important point, it is also important to consider that sexual violence may directly contribute to HIV transmission. The abstract would be more compelling if it started by acknowledging the complex ways in which violence, gender, and HIV are linked.

Thank you for this suggestion, we have revised the abstract accordingly.

Introduction

- Lines 40-42: The source of the definition for violence against children and youth is unclear. The citation provided is relevant for the annual estimation that over one billion children are affected by violence globally; however, the reference does not provide the quoted definition.

Thank you, we have provided an additional reference for the direct quote defining violence and have clarified that this is a World Health Organization definition. We’ve also added a citation with the definition of youth.

- Lines 44-46: It would strengthen the authors' introduction to amend this sentence to describe the wide-ranging consequences faced by individuals who experience violence during childhood per the Berenson, Wiemann, and McCombs article and other VAC-specific articles.

The sentence as currently written lays out many of the wide-ranging consequences described by Berenson et al. and some of the other cited work, e.g. poor physical and mental health, decreased education and employment opportunities, and increased health risk behaviors. We have further elaborated on the increased health risk behaviors (see track change manuscript lines 48-50) to highlight the negative effects of experiencing violence in youth.

Lines 46-47: The sentence that begins with "In addition to having the highest HIV prevalence..." starts quite abruptly. The authors may want to consider including a separate paragraph to describe the violence and HIV risks in sub-Saharan Africa or Uganda specifically.

Thank you, we have further elaborated on the risks of both HIV and violence in sub-Saharan Africa, and also a sentence on the prevalence on HIV and lifetime sexual violence in Uganda among females. Relevant citations have also been added.

- It is not always clear what differences exist when the authors mention children, youth, young adults, or adolescents. It would strengthen the readability if the authors used consistent terminology and provided corresponding definitions when utilizing various age-specific terminology.

Thank you, we have made an effort to use children and youth more consistently throughout the manuscript, with youth referring specifically to the 15-24 year old age group (see new UN citation for definition of youth provided - https://www.un.org/esa/socdev/documents/youth/fact-sheets/youth-definition.pdf).

Methodology

- The authors provide a clear description of the similarities between the VACS and UPHIA implementation and the authors guide readers to Tables 1 and 2 to better understand differences. It is important that the authors add a paragraph describing key differences as well.

Thank you, we have further elaborated on differences between the two surveys within a new paragraph in the methods section (lines 96-103 in track change version).

- When discussing the similarities between the VACS and UPHIA sampling strategies, it would be helpful to mention the split-stage design utilized by both.

Thank you, the revised paragraph explaining the differences between the surveys mentions this design used in both surveys.

- Table 2: There seem to be a few formatting inconsistencies or errors that the authors should review. For example, the narrative regarding repeated questioning per perpetrator is described differently in the first cell than in the second and third cell. 

Thank you, we have modified formatting for consistency in these cells.

As another example, the first example of "Physical Intimate Partner Violence (Past Year)" is an example of lifetime experience: "Has a romantic partner ever punched, kicked, whipped, or beat you with an object."

Thanks very much for pointing this one out, this was indeed an error that we have corrected.

Results

- Lines 106-112: The content in the first paragraph would benefit from a complementary Table.

We agree that a Table helps presents the data in a more digestible way, Table 3 has been added in the revised manuscript (see line 135 in track changed version). 

- Lines 106-113: It is unclear where there are statistically significant differences in findings provided in this paragraph, which is important since the authors describe differences and similarities between the two studies.

None of the differences presented in newly created Table 3 between UPHIA and VACS were statistically significant, either when comparing males or females. We have stated this in lines 134-135.

- Lines 108-109: It would help if the authors provided CIs rather than the IQRs

In Table 3, which was created based on the reviewers suggestion, we have added the 95% CI for ages.

Discussion

- Lines 141-143: The authors describe "significantly lower prevalence estimates of both lifetime physical and sexual violence for both males and females"; however, the findings related to sexual violence are mixed. For some questions, the UPHIA resulted in higher prevalence.

Thanks for pointing out some of the inconsistencies in the reporting here; we’ve modified the opening paragraph of the discussion to address both this comment and the following comment.

- Lines 142-143: The authors describe that past 12-month IPV was lower in the UPHIA than the VACS for females. It is unclear why the authors did not mention that past 12-month physical IPV was lower in the UPHIA than the VACS for males, as described in lines 116-117.

See comment above.

- Lines 143: Suggest that the authors use the term "more consistent" rather than "largest"

Thank you, modified as suggested.

- Lines 153-158: Sexual violence is often more sensitive to disclose; yet, there was not an overall lower reporting in the UPHIA than the VACS. The authors should clarify how interviewer selection and training may have influenced reporting for physical violence but not sexual violence.

Our results found that lower prevalence of sexual violence than physical violence in UPHIA than VACS, particularly in females; the difference was just not as great as the physical violence findings and were less consistent when looking at specific violence domains.

- Lines 159-172: The discussion would benefit from the authors further describing why they think the VACS consistently measured higher prevalence of physical violence but not sexual violence. Lines 168-172 indicate that the specificity of questions, including forms of violence and specific perpetrators, may be critical to increased reporting.

We agree that this is an interesting finding, and think that the specificity, skip patterns, and wording of the questions likely contributed to the larger difference between surveys in physical violence than sexual violence. We’ve elaborated on this point in line with the reviewer’s suggestion (lines 202-206 in track changed manuscript).

- When similar questions were asked in the VACS and UPHIA, such as sexual violence questions, results were more consistent between the two studies. Thus, the authors opinion that the PHIAs should not incorporate additional violence questions seems inconsistent with the evidence. However, the suggestion for the VACS to include biomarker data is compelling.

Thank you for making this important point; we have modified this paragraph in response to this comment as well as a comment made by another viewer. We agree with the reviewer that the more consistent estimates for questions asked in similar ways suggests that it may be possible to generate valid and reliable estimates of violence within an HIV-focused survey. However, this would likely involve expanding the questionnaire to be even longer, and given the differences in the primary objectives of the survey, the approach of having specialized surveys to attempt to estimate violence prevalence in children and youth along with HIV laboratory data makes the most sense. That way the survey can do a deeper dive into violence by asking each of the questions as intended. 

- Lines 184-185: The evidence provided in this manuscript suggests that the inclusion of perpetrators within the physical violence questions of the VACS enables it to better measure violence as a distal determinant of HIV acquisition, but the authors should clarify if/what other factors contribute to this conclusion.

Thanks for mentioning this – in addressing the previous comment and the other reviewer’s comments, we have reframed our concluding suggestions – see lines 218-227 in the revised track changed version of the manuscript.

---

## [Editor Report · Decision Letter 1]

22 Nov 2021

A comparison of two population-based household surveys in Uganda for assessment of violence against youth

PONE-D-21-25230R1

Dear Dr. Currie,

We’re pleased to inform you that your manuscript has been judged scientifically suitable for publication and will be formally accepted for publication once it meets all outstanding technical requirements.

Kind regards,

Lindsay Stark

Academic Editor

PLOS ONE
---

## [Editor Report · Acceptance letter]

2 Dec 2021

PONE-D-21-25230R1 

A comparison of two population-based household surveys in Uganda for assessment of violence against youth 

Dear Dr. Currie:

I'm pleased to inform you that your manuscript has been deemed suitable for publication in PLOS ONE. Congratulations! Your manuscript is now with our production department. 

Kind regards, 

on behalf of

Dr. Lindsay Stark 

Academic Editor

PLOS ONE